# Comparison between Quinoa and *Quillaja saponins* in the Formation, Stability and Digestibility of Astaxanthin-Canola Oil Emulsions

Daniela Sotomayor-Gerding [1],*[ID], Eduardo Morales [2] and Mónica Rubilar [2,3],*

1  Programa de Doctorado en Ciencias de Recursos Naturales, Universidad de La Frontera, Avenida Francisco Salazar 01145, Temuco 4811230, Chile
2  Scientific and Technological Bioresource Nucleus, BIOREN, Universidad de La Frontera, Avenida Francisco Salazar 01145, Temuco 4811230, Chile
3  Department of Chemical Engineering, Faculty of Engineering and Sciences, Universidad de La Frontera, Avenida Francisco Salazar 01145, Temuco 4811230, Chile
*  Correspondence: d.sotomayor02@ufromail.cl (D.S.-G.); monica.rubilar@ufrontera.cl (M.R.); Tel.: +56-45-2744232 (M.R.)

**Abstract:** Saponins from *Quillaja saponaria* and *Chenopodium quinoa* were evaluated as natural emulsifiers in the formation of astaxanthin enriched canola oil emulsions. The aim of this study was to define the processing conditions for developing emulsions and to evaluate their physical stability against environmental conditions: pH (2–10), temperature (20–50 °C), ionic strength (0–500 mM NaCl), and storage (35 days at 25 °C), as well as their performance in an in vitro digestion model. The emulsions were characterized, evaluating their mean particle size, polydispersity index (PDI), and zeta potential. Oil-in-water (O/W) emulsions were effectively produced using 1% oil phase and 1% emulsifier (saponins). Emulsions were stable over a wide range of pH values (4–10), but exhibited particle aggregation at lower pH, salt conditions, and high temperatures. The emulsion stability index (ESI) remained above 80% after 35 days of storage. The results of our study suggest that saponins can be an effective alternative to synthetic emulsifiers.

**Keywords:** oil-in-water (O/W) emulsion; emulsifier; saponin; astaxanthin; *Quillaja saponaria*; *Chenopodium quinoa*; canola oil; zeta potential; particle size; in vitro digestion





## 1. Introduction

Emulsifiers are surface active substances that facilitate emulsion formation and promote emulsion stability, affecting the particle size and the electrical repulsion between the particles [1]. Emulsifiers can be classified as synthetic, natural, finely dispersed solids, and auxiliary agents based on their chemical structure [2]. In recent years, the demand for healthier food products, containing more natural and environmentally friendly ingredients has increased, for which the use of natural emulsifiers has been the focus of recent research [3].

Currently, proteins, phospholipids, polysaccharides, lipopolysaccharides, bioemulsifiers (e.g., saponins, sophorolipids, rhamnolipids, and mannoproteins), and bioemulsifiers isolated from plant materials or produced by fermentation using bacteria, yeasts, or fungi (e.g., glycolipids, lipoproteins, and lipopeptides) are used as natural emulsifiers [2].

Due to their natural foam-like quality, the application of saponins as natural biosurfactants to improve the surface properties of food has recently been the subject of intensive study. Saponins are secondary metabolites mainly derived from plant materials [4]. These biosurfactants commonly contain a mixture of different amphiphilic constituents that have demonstrated their ability to form micelles when dispersed in water and support the formation and stabilization of oil-in-water emulsions. Their amphiphilic nature is given by the presence of hydrophilic regions (e.g., sugar groups) and hydrophobic regions

(e.g., phenolic groups) distributed within a single molecule [5]. Saponins have been found to have pharmaceutical properties of hemolytic, molluscicidal, anti-inflammatory, antioxidant, antifungal, antimicrobial, antiparasitic, antitumor, antiviral, and immune adjuvant activities [4,6]. Saponins may also be effective at inhibiting lipid oxidation in emulsions because of their radical-scavenging capacity [7].

Saponins from *Quillaja saponaria* have been used for the preparation of oil-in-water (O/W) emulsions in several studies, using medium chain triglycerides (MCT) [8,9], orange oil [10], and for the encapsulation of vitamin E [11]. Furthermore, the use of saponins from *Q. saponaria* mixed with other surfactants such as sodium caseinate, pea protein, rapeseed lecithin, egg lecithin [12], Tween 80 [13], β-lactoglobulin [14], or hydrolyzed rice glutelin [15] have been reported in the literature.

The presence of saponins has been also reported in quinoa (*Chenopodium quinoa*) [16,17]. Several studies have reported the use of *C. quinoa* extracts as emulsifier. Authors have reported the use of quinoa starch for the preparation of Pickering emulsions [18–20] and the use of protein isolated for the preparation of emulsion gels [21] and high internal phase emulsions [22–24]. However, the use of quinoa saponins in the preparation of emulsions has not been reported in the literature. On the other hand, studies have highlighted the health benefits of quinoa derived products [25] and a recent study reported a safety assessment for the oral use of saponins from *C. quinoa* in rats reporting no adverse effects under a dose of 50 mg/kg/day [26].

Although there are a wide variety of studies on the use of saponins as emulsifiers, the incorporation of bioactive ingredients has not been extensively studied. The incorporation of carotenoids such as astaxanthin in emulsions is of great interest in the food industry, as this is a pigmented compound with many health benefits [27]. Nevertheless, their utilization as nutraceutical ingredients within foods is currently limited because of their poor water-solubility, high melting point, chemical instability, and low bioavailability [28].

Consequently, the aim of this study was to evaluate saponins from *Quillaja saponaria* and *Chenopodium quinoa* as natural emulsifiers in the formation and stabilization of astaxanthin-enriched canola oil emulsions. The performance of these extracts was compared to that of a synthetic surfactant (Tween 20) that is currently widely used in the food and beverage industry to formulate emulsion-based products. The influence of environmental stresses (pH, ionic strength, and temperature) and storage on the stability of the resulting emulsions against droplet growth and gravitational separation was evaluated and the in vitro digestion was also investigated to provide information on their gastrointestinal transformation and/or absorption.

## 2. Materials and Methods

### 2.1. Materials

The extracts from *Q. saponaria* (190 g/L saponin) and *C. quinoa* were provided from South extracts S.A. (Perquenco, Chile), canola oil was purchased from a local market and astaxanthin oleoresin (Supreme Asta oil 5.0%) from Atacama Bio Natural Products S.A. (Iquique, Chile). Distilled water used in this study had a conductivity 0.90 μS/cm.

Non-ionic surfactant polyoxyethylene (20) sorbitan monolaurate (Tween 20; P1379), sodium chloride (NaCl; 746398), sodium hydroxide (NaOH; S5881), dipotassium hydrogen phosphate ($K_2HPO_4$; P3786), mucin from porcine stomach Type II (M2378), pepsin from porcine gastric mucosa (P7012), bile extract porcine (B8631), and pancreatin from porcine pancreas (P1750) were purchased from Sigma Aldrich Co. (St. Louis, MO, USA). All other chemicals were of analytical grade.

The *Q. saponaria* and *C. quinoa* extracts were characterized according to the AOAC methods [29], measuring their dry weight, refractive index, and solids percent in a refractometer (Abbe Mark II plus, Reichert Inc., Depew NY, USA) and pH.

## 2.2. Emulsion Formation and Characterization

O/W emulsions were prepared using 99% aqueous phase and 1% oil phase. The aqueous phase was obtained, dispersing 1% emulsifier (*Q. saponaria*, *C. quinoa* or Tween 20) in distilled water, and the oil phase was prepared by mixing astaxanthin (2g/L) with canola oil (1:1). The emulsions were homogenized (5000 rpm, 10 min, Pro400DS benchtop homogenizer, Pro Scientific Inc., Oxford, CT, USA) and subsequently passed through the high-pressure homogenizer (4 cycles, 100 MPa, PandaPlus 2000, GEA Niro Soavi, Parma, Italy).

The average particle size and polydispersity index (PDI) of emulsions were determined by dynamic light scattering and the surface charge (zeta potential) by electrophoretic mobility in a Zetasizer (Nano-ZS90, Malvern Instruments, Worcestershire, UK). Measurements were performed on diluted (1:100 distilled water) emulsions.

The influence of emulsifier percentage on the mean particle size and zeta potential of emulsions was evaluated using 0.1, 0.5, 1, 2, and 5% of emulsifier.

## 2.3. Influence of Environmental Changes on Emulsion Physical Stability

The influence of different environmental conditions that might be encountered in the processing of food on the emulsion stability were evaluated. Emulsions were prepared using 1% oil phase and 1% emulsifier. The effect of pH on emulsion stability was evaluated by manually adjusting the pH of emulsions at 1 unit interval from 2 to 10, after dilution (1:100), emulsions were evaluated in a Zetasizer, measuring the mean particle size and zeta potential. The effect of temperature on emulsion stability was investigated by measuring the average droplet size and zeta potential using a step-wise protocol, in which the temperature was changed in steps of 5 °C from 20 to 50 °C. Temperature was stabilized with a Peltier temperature control of the Zetasizer equipment. The influence of ionic strength on emulsion stability was determined by adjusting the salt concentration to between 0 and 500 mM NaCl prior to dilution (1:100) and analysis with the Zetasizer at 25 °C. Representative photographs of the emulsions were taken after 24 h of incubation with different conditions of pH (2–10) and salinity (0–500 mM NaCl).

The emulsion stability index (ESI) was determined by monitoring the extent of gravitational phase separation during storage for 35 days at 25 °C in darkness according to previous reports [30].

## 2.4. In Vitro Digestion of Emulsions

An in vitro gastrointestinal tract (GIT) model was used to simulate mouth, gastric, and small intestine digestion according to our previous report [31]. Freshly prepared emulsions were mixed (1:1) with simulated saliva fluid (SSF) containing mucin (5 g/L). The pH of the mixture was adjusted to 6.8 prior to incubation at 37 °C for 10 min with continuous agitation at 100 rpm to simulate the mouth phase. For the gastric phase, simulated gastric fluid (SGF) was prepared by dissolving NaCl (2 g) and HCl (7 mL) in a liter of water adjusted to pH 1.2. The previously processed emulsion was mixed with SGF (1:1, *v/v*) and pH adjusted to 1.5 prior to incubation at 37 °C for 10 min with continuous agitation at 100 rpm. Pepsin (3 mg) was added to the mixture after 10 min and samples were incubated for 2 h with the previous conditions (37 °C, 100 rpm). For the intestinal phase, simulated intestinal fluid (SIF) containing $K_2HPO_4$ (6.8 g/L), 0.2 M NaOH (190 mL/L), and maintained at pH 7.5 was mixed with the samples from the gastric phase (1:3, *v/v*, for a total of 30 mL) and bile extract (0.15 g). This mixture was maintained at 37 °C after adjusting the pH to 7. The small intestinal phase was simulated with a pH-stat (Metrohm USA Inc., Riverview, FL, USA) to maintain constant pH (7) of the solution by adding 0.05 M NaOH solution. The volume of NaOH required to neutralize the free fatty acids (FFA) was recorded for 20 min. Once the equipment was prepared, freshly prepared pancreatin suspension (2.5 mL; 24 mg/mL)

dissolved in phosphate buffer was added to the mixture, to initiate the reaction. The amount of free fatty acids released from lipid digestion was calculated as follows:

$$\text{FFA(mM)} = (V_{\text{NaOH T}} - V_{\text{NaOH T0}}) \times M_{\text{NaOH}} \times 1000$$

Here, $V_{\text{NaOHT}}$ is the volume (L) of sodium hydroxide required to neutralize the FFA produced, $V_{\text{NaOH T0}}$ is the volume (L) of sodium hydroxide added at the beginning of the reaction, and $M_{\text{NaOH}}$ is the molarity (M) of the sodium hydroxide solution used.

### 2.5. Statistical Analysis

All measurements were performed in triplicate and results were expressed as the mean and the standard deviation. All the results of this study were subjected to one-way analysis of variance (ANOVA). Significant differences ($p \leq 0.05$) between means were determined by Tukey's tests.

## 3. Results

### 3.1. Emulsion Formation and Characterization

*Q. saponaria* and *C. quinoa* extracts were characterized (Table 1). The extracts had similar characteristics, a solid concentration of ~21%, a refractive index of 1.36, and a humidity of ~80%. The only characteristic that was a little different between the extracts was the pH. *Q. saponaria* extracts had a pH of 3.94, and the *C. quinoa* extract had a slightly lower pH of 3.49.

**Table 1.** Characteristics of *Quillaja saponaria* and *Chenopodium quinoa* extracts.

| Characteristic | *Quillaja saponaria* | *Chenopodium quinoa* |
|---|---|---|
| % Solids (°Brix-TC [1]) | 20.83 ± 0.060 | 20.70 ± 0.200 |
| Refractive Index | 1.36 ± 0.000 | 1.36 ± 0.000 |
| pH | 3.94 ± 0.010 | 3.49 ± 0.010 |
| Humidity (%) | 79.39 ± 0.005 | 81.22 ± 0.140 |

[1] TC: Temperature compensated.

Emulsions were effectively produced using saponins from *Q. saponaria* and *C. quinoa* (Table 2). Tween 20 emulsions were produced as a control for comparison. *Q. saponaria* emulsions had the smallest mean particle size, 189 nm, while *C. quinoa* emulsions had a mean particle size of 316 nm. Emulsions prepared under the same conditions using Tween 20 had a mean particle size of 205 nm. Polydispersity index was similar between the emulsions, 0.32, 0.33, and 0.35, for emulsions prepared with *Q. saponaria*, *C. quinoa*, or Tween 20 as emulsifier, respectively.

**Table 2.** Mean particle size, zeta potential and polydispersity index (PDI) of emulsions prepared with *Q. saponaria* or *C. quinoa* extracts as emulsifier.

| Emulsifier | Mean Particle Size (nm) | PDI | Zeta Potential (mV) |
|---|---|---|---|
| *Q. saponaria* | 189 ± 5 a | 0.32 ± 0.005 a | −29.6 ± 0.3 a |
| *C. quinoa* | 316 ± 8 b | 0.33 ± 0.003 a | −27.7 ± 1.1 ab |
| Tween 20 | 205 ± 10 a | 0.35 ± 0.014 a | −26.0 ± 1.2 b |

Different lowercase letters in a column indicate significant differences ($p \leq 0.05$) among the different experimental groups (*Q. saponaria*, *C. quinoa*, Tween 20).

Regarding the zeta potential values, all emulsions presented highly negative surface charges. *Q. saponaria* emulsions had a zeta potential of −29.6 mV, *C. quinoa* emulsions had a less negative charge of −27.7 mV and Tween 20 emulsions had a zeta potential of −26.0 mV. Statistical analysis showed that emulsions prepared with *C. quinoa* saponins had significantly higher particle size.

The influence of emulsifier concentration was assessed (Figure 1), evaluating five emulsifier percentages (0.1, 0.5, 1, 2, and 5%). For *Q. saponaria* emulsions, the average particle size decreased from 329 nm to 189 nm when the emulsifier concentration increased from 0.1 to 1%; however, the average particle size increased from 189 nm to 348 nm when the emulsifier concentration increased from 1 to 5%. The smallest mean particle size (189 nm, PDI: 0.32) was obtained using 1% emulsifier and 1% oil phase. *C. quinoa* emulsions had a similar mean particle size (~330 nm) between 0.1 and 1% emulsifier, statistical analysis did not show significant differences within this range. However, a significant increase in particle size was observed using 2 and 5% emulsifier (Figure 1a). Using 2% *C. quinoa* saponins as emulsifier, an average particle size of 931 nm and a PDI of 0.61 were obtained, while using 5% emulsifier an average size of 3893 nm and a PDI of 1 were obtained. The smallest particle size (316 nm, PDI: 0.33) was obtained using a 1% concentration of *C. quinoa* emulsifier.

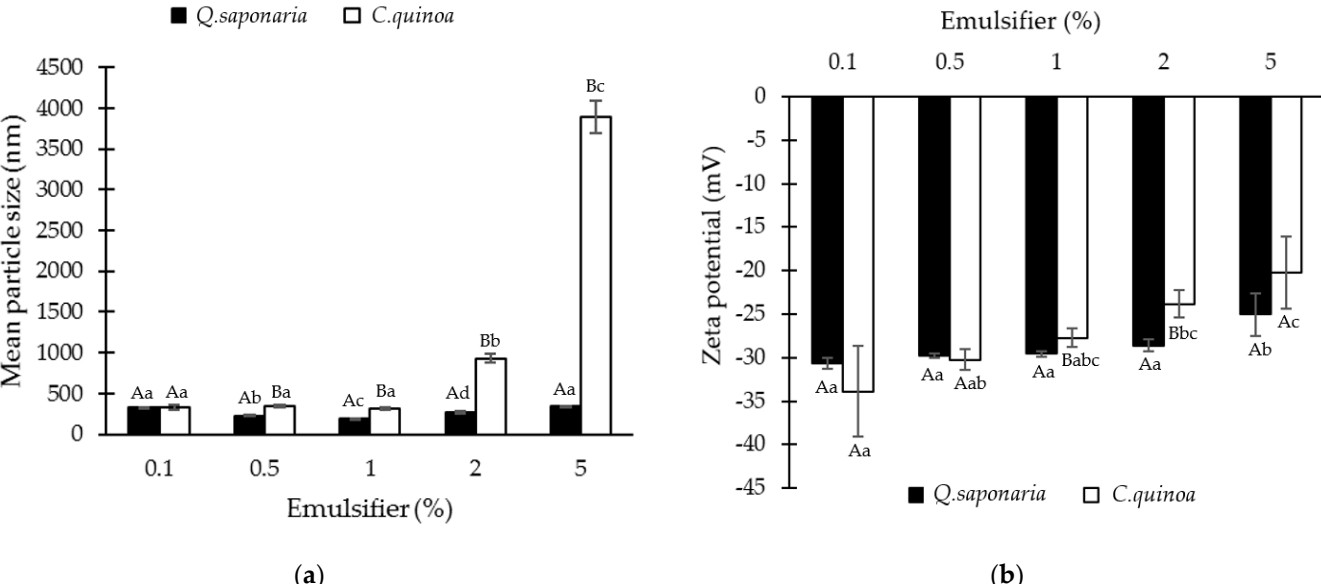

**Figure 1.** Influence of emulsifier concentration on emulsion (**a**) mean particle size and (**b**) zeta potential. Different capital letters indicate significant differences ($p \leq 0.05$) of the mean particle size or zeta potential among the different emulsion types (*C. quinoa*, *Q. saponaria*) within the same emulsifier concentration. Different lowercase letters indicate significant differences ($p \leq 0.05$) of the mean particle size or zeta potential among the different emulsifier concentrations (0.1, 0.5, 1, 2, 5%) within the same emulsion type.

Regarding the zeta potential, the effect of emulsifier concentration had the same tendency for emulsions generated with saponins of *Q. saponaria* and *C. quinoa*, the zeta potential increased as the emulsifier percent increased (Figure 1b), obtaining a less negative charge. Statistically significant differences between *Q. saponaria* and *C. quinoa* emulsions were obtained using 1 and 2% of emulsifier. The highest zeta potential value was −20.3 mV for *C. quinoa* emulsions with a 5% of emulsifier.

The 1% emulsifier concentration was used in the following evaluations, considering these results, small particle size, and zeta potential above ±20 mV.

### 3.2. Influence of Environmental Changes on Emulsion Physical Stability

Emulsions were physically stable over a wide range of pH values (4–10), the mean particle size had no significant changes between this range (Figure 2a), which can also be observed in Figure 3 where no phase separation, creaming, or other form of destabilization of the emulsion is observed. In addition, an increase in the negative charge of the particles was observed (Figure 2b) for all emulsions. For example, *Q. saponaria* zeta potential changed from −28.3 mV at pH 4 to −39.3 mV at pH 10. However, *Q. saponaria* and

Tween 20 emulsions were destabilized at pH 2, while *C. quinoa* emulsions were destabilized at pH 2 and 3 (Figure 2a).

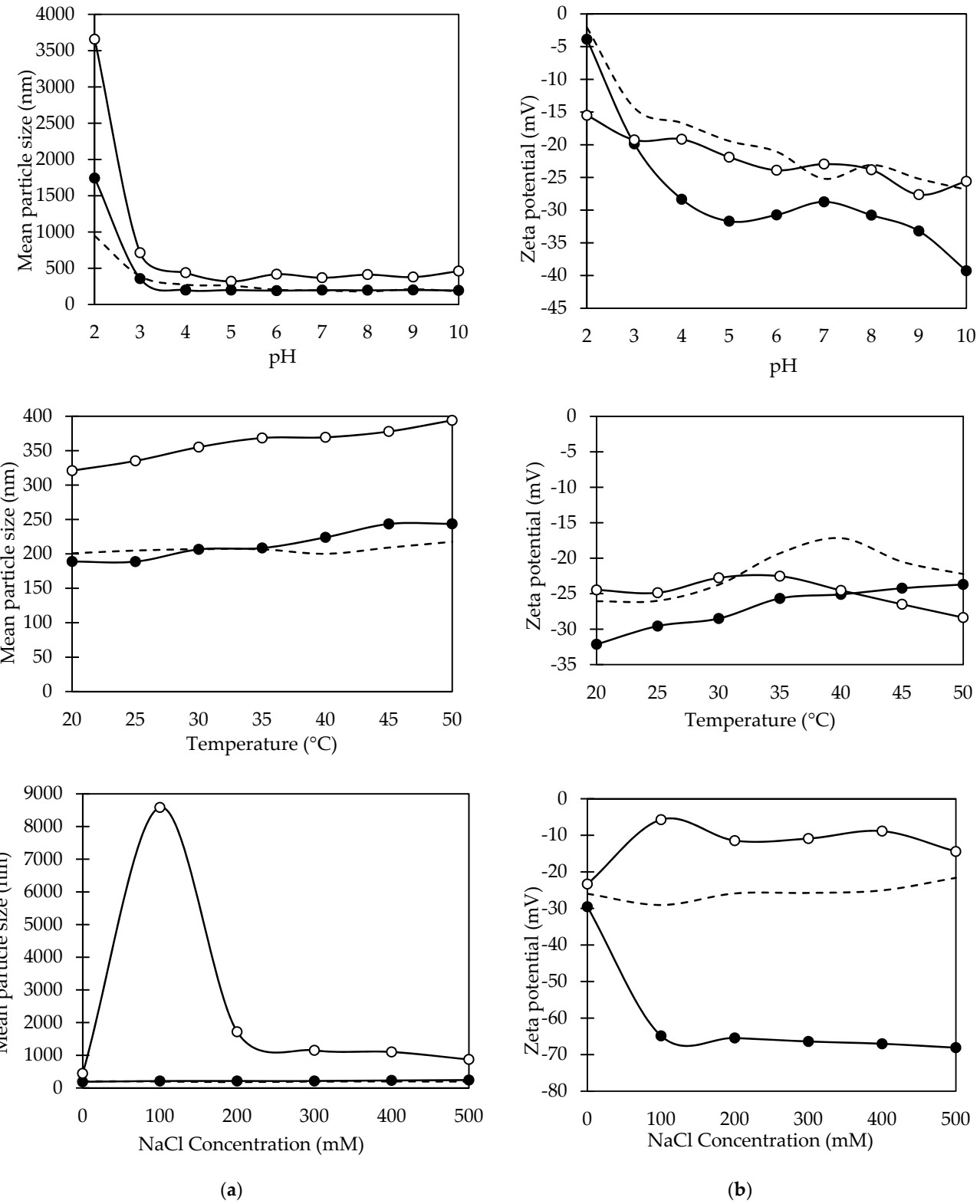

(**a**)  (**b**)

**Figure 2.** Influence of environmental changes: temperature (20–50 °C), pH (2–10) and ionic strength (0–500 mM) on emulsion stability expressed as changes in (**a**) mean particle size and (**b**) zeta potential for emulsions produced with: (●) *Quillaja saponaria*, (○) *Chenopodium quinoa* or (–) Tween 20 as emulsifier. Emulsions were prepared using 1% oil phase and 1% emulsifier.

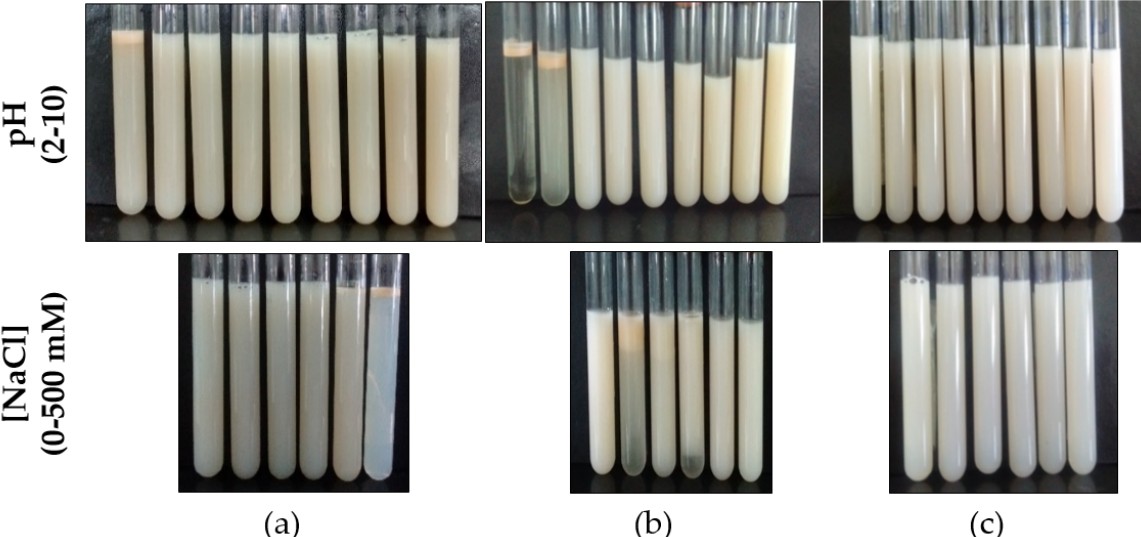

**Figure 3.** Photographs of emulsions prepared with (**a**) *Quillaja saponaria*, (**b**) *Chenopodium quinoa*, or (**c**) Tween 20 after incubation for 24 h with different environmental conditions: pH (2–10) or ionic strength (0–500 mM NaCl).

The mean particle size increased significantly at pH 2 for all emulsions, *C. quinoa* emulsions reached a mean particle size of 3656 nm, *Q. saponaria* 1741 nm and Tween 20,946 nm. Particle aggregation and emulsion destabilization can also be observed in Figure 3, a creaming line is produced for emulsions generated with *Q. saponaria* at pH 2 and for emulsions generated with *C. quinoa* at pH 2 and 3. Emulsion destabilization can be correlated to the reduction of the negative charge of the particles. The zeta potential of the particles increased significantly at acid pH. At pH 2, *Q. saponaria* emulsions had a zeta potential of −3.9 mV, *C. quinoa* emulsions showed a zeta potential of −15.5 mV and Tween 20 emulsions had a zeta potential of −2.1 mV.

The effect of temperature on emulsions prepared with different emulsifiers was evaluated by measuring the change in the mean particle size (PS) and zeta potential (ZP). The changes on temperature led to some fluctuations in the mean particle size and zeta potential. For emulsions prepared with *Q. saponaria* and *C. quinoa*, the particle size increased linearly with increasing temperature, but at a different rate for each emulsion (Figure 2). The equations for emulsions, *Q. saponaria* and *C. quinoa* were: PS = 2.08 · [T°] + 142.08, $R^2$ = 0.9473; PS = 2.28 · [T°] + 280.44, $R^2$ = 0.9555, respectively. For *Q. saponaria* emulsions the mean particle size increased significantly over 40 °C. For emulsions prepared with Tween 20, the size remained relatively stable, with a standard deviation between samples of only 6 nm, however, a linear regression did not provide a good fit ($R^2$ < 0.5), given the fluctuations in size.

The zeta potential of emulsions prepared with *Q. saponaria* extract as emulsifier increased linearly with increasing temperature (ZP = 0.2812 · [T°] − 36.827, $R^2$ = 0.9362). For emulsions prepared with *C. quinoa* extract and Tween 20 as emulsifier, the correlation between zeta potential and temperature was not clear. For Tween 20 emulsions, the lowest negative charge (−17.2 mV) was observed at 40 °C, in the case of emulsions with *C. quinoa*, the lowest negative charge was observed at 35 °C, obtaining a zeta potential of −22.5 mV.

The presence of salts in the aqueous phase significantly affected the stability of the emulsions. A linear increase in the mean particle size was observed for *Q. saponaria* emulsions (PS = 0.0911 · [NaCl] + 193.84, $R^2$ = 0.893), emulsion destabilization is observed at 500 mM NaCl (Figure 3a). *C. quinoa* emulsions proved to be more sensitive to the influence of salinity, observing destabilization at 100 mM NaCl, where a mean particle size of 8584 ± 672 nm was determined (Figure 2a) and the PDI reached the value of 1. Between 200 mM NaCl and 500 mM NaCl the mean particle size was 3 times higher than the

emulsion at its normal state and the PDI values were above 0.59. Emulsion destabilization effect can be observed in Figure 3b. Tween 20 emulsions remained stable at different concentrations of NaCl (Figure 3c), a slight decrease in the mean particle size was observed, varying from 205 nm at 0 mM NaCl to 194 nm at 500 mM NaCl.

Regarding zeta potential, *Q. saponaria* emulsions showed a significant increase in the negative charge of the particles with increasing ionic strength. At 0 mM NaCl concentration, *Q. saponaria* emulsions had a zeta potential of −29.6 mV, while emulsions exposed to higher concentrations of NaCl (100–500 mM) had a zeta potential that varied between −64.9 and −68.1 mV. For *C. quinoa* emulsions, the zeta potential reacted in the opposite way, the negative charge was reduced, at 100 mM the highest zeta potential of −5.7 mV was determined, between 200 and 500 mM NaCl zeta potential remained close to −10 mV. In the case of Tween 20 emulsions, the negative charge remained stable, decreasing slightly as the NaCl concentration increased.

The emulsion stability index (ESI) was determined, monitoring the extent of gravitational phase separation for 35 days (Figure 4).

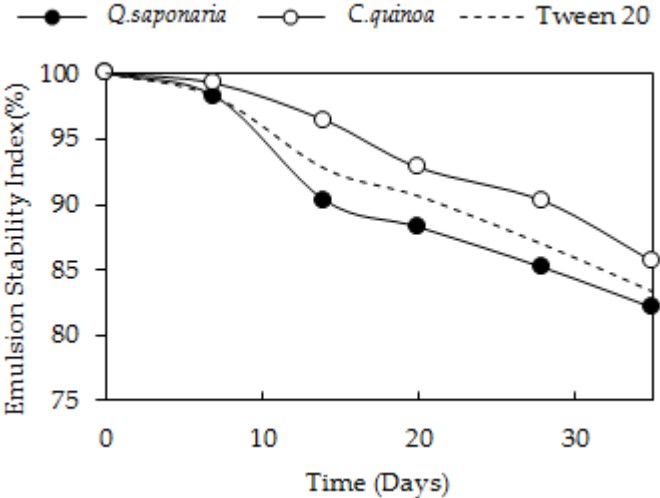

**Figure 4.** Emulsion stability index (ESI) after 35 days of storage at 25 °C for emulsions prepared with (●) *Quillaja saponaria*, (○) *Chenopodium quinoa* or (–) Tween 20 as emulsifier.

All emulsions were stable and homogeneous immediately after preparation, starting with an ESI of 100%. Phase separation increased with storage time, revealing the creaming of emulsions. ESI decreased with storage at slightly different rates depending on the emulsifier used, represented by the equations: ESI = −0.5363 · [time] + 99.935, $R^2$ = 0.9585, for *Q. saponaria*; ESI = −0.4192 · [time] + 101.33, $R^2$ = 0.9685, for *C. quinoa* and ESI = −0.4904 · [time] + 100.44 $R^2$ = 0.9885, for Tween 20. At day 35 the ESI values were 82.04 ± 1.13%, 85.64 ± 0.89%, and 83.27 ± 4.33% for emulsions prepared with *Q. saponaria*, *C. quinoa*, or Tween 20, respectively. No significant differences were found between emulsions at day 35 of storage, indicating that the emulsifier type did not affect their stability during storage time.

*3.3. In Vitro Digestion of Emulsions*

Emulsions remained stable during the mouth and stomach phases, and the release of fatty acids occurred at intestinal level where their absorption usually takes place. After approximately seven minutes of reaction, all fatty acids were released. The curves of the fatty acid release were very similar for the different types of emulsifier (Figure 5). *C. quinoa* and Tween 20 emulsions reached a similar concentration of fatty acids released, 87.6 mM and 90.6 mM, respectively. However, *Q. saponaria* emulsions achieved a slightly higher amount of fatty acid released, 101.7 mM. Statistical analysis indicates that there are no

significant differences between the curves, which suggests that the emulsifier type does not play an important role in this assay.

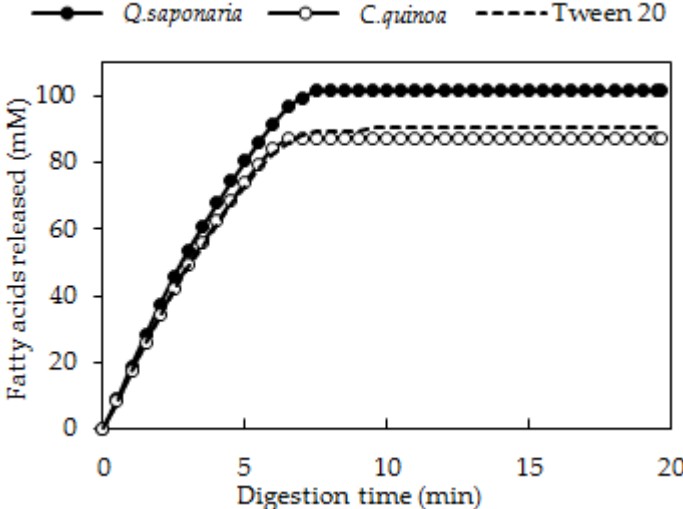

**Figure 5.** Fatty acids release curves generated during intestinal digestion for emulsions generated with (●) *Quillaja saponaria*, (○) *Chenopodium quinoa*, or (–) Tween 20 as emulsifier.

## 4. Discussion

Saponins from *Q. saponaria* and *C. quinoa* were effective for the formation of astaxanthin enriched oil-in-water emulsions. The smallest mean particle size was obtained using *Q. saponaria* saponins. According to previous studies, saponins from *Q. saponaria* are usually effective at forming small droplets, due to their relatively low molecular weight (~1.67 kDa) they tend to form thin interfacial layers [3]. However, for emulsions prepared with *C. quinoa* saponins, a significantly higher size was obtained. In general, a small particle size is sought since they can improve the system, having a better stability and bioaccessibility after ingestion [31]. All emulsions presented highly negative surface charges, ranging from −29.6 to −26.0 mV. According to previous studies [32], the magnitude of zeta potential gives an indication of the potential stability of the colloidal system and generally, values greater than ±20 mV produce systems that are stable over time.

After evaluating the effectiveness of the extracts to form emulsions, the effect of different proportions of the emulsifier was evaluated. Studies forming oil-in-water emulsions with saponins use emulsifier percentages ranging from 0.001% to 2% [8,10,11]. In our study, the range of 0.1 to 5% was evaluated.

Previous studies have reported a dependence between emulsifier concentration and particle size, where droplet size decreases with increasing emulsifier concentration [32]. For oil-in-water emulsions containing heavy crude, droplet size decreased with increasing Tween 20 concentrations from 0.1 to 2.1 wt% [33]. Similarly, in the production of food grade Pickering emulsions average particle size decreased from ~25 to ~0.15 μm as the concentration of Tween 20 increased from 1 to 4% [34]. Authors attributed this effect to (i) a higher surfactant concentration means that a larger surface area can be stabilized during homogenization; (ii) a higher surfactant concentration leads to faster coverage of the droplet surfaces by surfactant molecules, and therefore better protection against recoalescence [35]. Our study coincided with the tendency reported by other authors in the range between 0.1 to 1%, however, above 2% an increase in the obtained particle size was observed. For *Q. saponaria* emulsions, the average particle size increased 1.84-fold between 1% and 5% emulsifier concentration, and the PDI increased to 0.4. For *C. quinoa* emulsions, the effect was more significant, between 1 and 2% of emulsifier concentration, the average particle size increased 2.95-fold and the PDI increased to 0.6; between 1 and 5% of emulsifier concentration, the average particle size increased approximately 12-fold and the PDI reached the value of 1, indicating that the sample is highly polydisperse with

multiple size populations. This destabilization of the emulsion could be an effect of the reduction of the negative charge of the particles by increasing the concentration of the emulsifier. When using 5% of *C. quinoa* saponins as emulsifier, the limit of the stable zone for colloidal particles is reached. For electrostatically-stabilized emulsions, the magnitude of the zeta potential should be greater than about 20 mV to produce systems that are stable during long-term storage [32]. It should be considered that the extracts from *Q. saponaria* and *C. quinoa* are compositionally complex materials that will contain a variety of surface active components with different surface activities, for *Q. saponaria* around 100 saponins have been reported, where the majority of these consist of quillaic acid substituted with oligosaccharides at C-3 and C-28 [36], while for *C. quinoa* the main sapogenins reported are oleanolic acid, hederagenin, and phytolaccagenin [17], consequently, it is possible that a portion of the components of these extracts may affect the stability of the emulsion.

Accordingly, the use of these extracts as an emulsifier should be kept in the range of 0.1 to 1% to ensure the formation and stability of the emulsion.

Emulsions may become unstable through a number of different instability mechanisms (e.g., flocculation, coalescence, Ostwald ripening, and gravitational separation), which depend on storage conditions such as pH, ionic strength, and temperature [37]. We therefore examined the influence of different environmental conditions that might be encountered in the processing of food on the stability of emulsions.

The acid pH had a strong effect on the stability of emulsions, zeta potential values increased significantly, and particle size increased generating a cream layer at pH 2 for *Q. saponaria* emulsions, and at pH 2 and 3 for *C. quinoa* emulsions. Previous studies have shown that changes in pH can have an immediate and significant effect on the zeta potential of emulsions [38,39]. An explication to this effect is that the high concentration of protons present in the aqueous phase at acidic pH neutralize the negative initial charges of the particles. Therefore, the increase in the mean particle size could be a secondary effect of lowering the pH, i.e., at lower pH, the charge of the particles decreases, and hence the electrostatic repulsion becomes insufficient, causing the phenomenon of aggregation observed. This correlation between the reduction in absolute zeta potential and the increase in particle size has been reported previously [39].

Previous studies report a similar trend regarding the zeta potential, where, as the pH increases, the zeta potential becomes more negative, and as the pH decreases, the negative charge is reduced, explaining the effect on the adsorption of hydrogen $H^+$ and hydroxyl $OH^-$ ions [40]. Saponin emulsions, especially emulsions prepared with *C. quinoa* were more susceptible to the effect of pH, unlike the emulsions prepared with Tween 20 in which the creaming process was not observed. Furthermore, studies suggest that acidic pH values could be unfavorable for emulsions containing carotenoids, since the rate of degradation of the carotenoid at acidic pH is higher [41]. Consequently, the acid pH would not be favorable for emulsions containing carotenoids due to their degradation and destabilization.

The effect of temperature was also evaluated, observing an increase in particle size, especially in emulsions prepared with saponins. The increase on the mean particle size might be related to the Brownian motion of the particles. Considering that higher temperatures increase the movement of particles and hence increase the collisions between the particles, which could generate coalescence or aggregation phenomena [32]. The emulsions prepared with Tween 20 remained relatively stable when facing different temperatures, having a standard deviation between the samples of only 6 nm. Although linear regression did not provide a good fit ($R^2$ = 0.4723), it can be seen that there are no significant changes in the average particle size.

The ionic strength of emulsified foods may vary considerably depending on the nature of the food products in which the oil droplets are present. Consequently, the effect of ionic strength on the emulsion stability was evaluated. Studies suggest that high concentration of salts destabilize emulsions generating particle aggregation [37]. The phenomenon where high concentration of ions in the aqueous phase (produced by the dissociation of salts) invalidate the repulsive charges between the particles is known as "electrostatic screening" [42].

In the case of *Q. saponaria* emulsions, an adsorption of anions from the aqueous phase is presumed, which is observed in the significant decrease in the zeta potential, generating a destabilization of the system at high NaCl concentrations. *For C. quinoa,* an unusual effect was observed where the greatest destabilization was generated at a concentration of 100 mM NaCl, the negative charge was reduced, the particle size increased almost 20 times, and the polydispersity index reached the value of 1, generating a thick line of creaming, which can be observed in Figure 3b. Between 200 and 500 mM NaCl, an increase in the average particle size and emulsion destabilization were also observed, however, it was not as significant as previously observed. In the case of Tween 20 emulsions, significant changes were determined, however, they did not generate creaming by particle aggregation. The size-enhancing effect has been previously reported in n-alkane emulsions where an increase from 450 nm to 1300 nm in the presence of NaCl is described [38].

The effect of storage time on emulsion stability was evaluated. After 7 days, the gravitational separation of phases begins to be observed, the ESI values are reduced to ~85% at 35 days of evaluation. In emulsion with small droplets, this phenomenon is expected, given that they are thermodynamically unstable systems. In nanoemulsions, Brownian motion dominates the movement of the particles and destabilization is caused by Ostwald ripening [43].

Finally, the release of fatty acids was evaluated after in vitro digestion, release curves were very similar; however, *Q. saponaria* emulsions achieved a higher amount of fatty acids released. The slight differences between the curves might be given by the initial different mean particle sizes between the types of emulsions [31].

## 5. Conclusions

Our study demonstrates that oil-in-water astaxanthin enriched emulsions can be effectively produced using a 1% oil phase and 1% saponins from *Q. saponaria* or *C. quinoa* extract as emulsifier. The average particle size depended on the emulsifier used, the smallest particle size was obtained with saponins from *Q. saponaria* and the largest particle size with saponins from *C. quinoa.* Additionally, it was determined that the concentration of saponins as emulsifier significantly affected the particle size and the zeta potential obtained. Emulsions with a size smaller than 350 nm could be obtained using *Q. saponaria* saponin concentrations between 0.1 and 5%, and *C. quinoa* saponin concentrations between 0.1 and 1%. Concentrations of *C. quinoa* saponins between 2 and 5% generated destabilization of the emulsion.

The emulsions had slightly different responses to the effect of environmental conditions. Tween 20 emulsions were stable over a wide range of pH values (3–8), salt concentrations (0–500 mM NaCl), and temperatures (20–50 °C); *Q. saponaria* emulsions were unstable at low pH values (2), high NaCl concentrations, and high temperatures (over 40 °C) and *C. quinoa* emulsions were highly unstable to droplet aggregation and phase separation at low pH values (2–3) and moderate ionic strengths (>100 mM NaCl). Emulsions remained stable during in vitro digestion, releasing fatty acids at intestinal level. *C. quinoa* emulsions released fatty acids at the same level as Tween 20 emulsions; however, *Q. saponaria* emulsions achieved a slightly higher amount of released fatty acids.

The results of our study contribute to increase the knowledge about the use of saponins from different natural sources in the formation of oil-in-water emulsions, and suggest that saponins can be an effective alternative to synthetic emulsifiers and even superior in terms of releasing bioactive compounds.

**Author Contributions:** Conceptualization, D.S.-G. and M.R.; methodology, D.S.-G.; validation, D.S.-G.; formal analysis, D.S.-G. and E.M.; investigation, D.S.-G.; resources, M.R.; data curation, D.S.-G.; writing—original draft preparation, D.S.-G.; writing—review and editing, D.S.-G., E.M. and M.R.; visualization, D.S.-G.; supervision, M.R.; project administration, M.R.; funding acquisition, D.S.-G. and M.R. All authors have read and agreed to the published version of the manuscript.

**Funding:** This research was funded by ANID through Doctoral Scholarship n° 21150735 and through FONDECYT project n° 1160558.

**Institutional Review Board Statement:** Not applicable.

**Informed Consent Statement:** Not applicable.

**Data Availability Statement:** Not applicable.

**Acknowledgments:** The authors would also like to acknowledge the support of the Scientific and Technological Bioresource Nucleus (BIOREN) at the Universidad de La Frontera for granting access to their equipment. Special thanks to Reinaldo Briones from South Extracts company that supplied the *Quillaja saponaria* and *Chenopodium quinoa* extracts.

**Conflicts of Interest:** The authors declare no conflict of interest. The funders had no role in the design of the study; in the collection, analyses, or interpretation of data; in the writing of the manuscript; or in the decision to publish the results.

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
