# Peer review of "Comparison between Quinoa and Quillaja saponins in the Formation, Stability and Digestibility of Astaxanthin-Canola Oil Emulsions"

_colloids, doi:10.3390/colloids6030043_

Round 1

Reviewer 1 Report

Referee report Colloids and Interfaces

 Comparison between Quinoa and Quillaja saponins in the formation, stability and digestibility of astaxanthin- canola oil emulsions

Daniela Sotomayor- Gerding, Eduardo Morales and Mónica Rubilar

Dear Editor,

This paper presents interesting results and nowadays very important which can be useful for different kind of industries but the way of presentation should be better. This paper can be recommended to publish after small revision and following changes should be done before the paper acceptance.

General remark:

Fragments requiring corrections are marked in colour in the main text.

Abstract

Line 22

From what level decreased to 85%?

Keywords

Should be more informative

 Introduction

Line 52-53

Whole sentence, English should be corrected.

Line 55, 56

Wrong names

Materials

Line 94

Should be added purity of water, conductivity?

Line 102

Dot at the end of the sentence

Line 124 onwards

Purity of components should be added not only of the company, the presentation is unprofessional

At minute 10???? Please correct

According SI units symbol of liter should be capital letter L, previously in the text it was correct

Line 145-147

The designations of symbols in the equation in my opinion are unfortunate, please introduce any other, more informative?.

Line 174 onwards

Use the same precision range for data, e.g. for zeta potential to one decimal place. Smaller or larger values ​​are confusing for negative potential values, it is safer to use absolute zeta potential values ​​when discussing the results.

Line 177

Polydispersity, we write this word together

Line 183

This wording is not mathematically correct.

Line 184 onwards

These potential values ​​seem to be confusing to the potential reader, because they do not match those in the table, please specify.

Line 192

In this sentence it is about negative zeta potential values, or absolute values?, because in my opinion this trend should be the opposite. Please check?

Line 207

This word “cremation” is unfortunate, because meaning as a result of this process we have a body cremated just like at a funeral. It should be change for “creaming”, because this is correct naming of this chemical process during the emulsion aging.

Results

A deeper analysis is needed and a comparison with the work of other scientists. Please refer in the introductory part and/or in the discussion the analogous results.   For example:

Effect of ionic strength on electrokinetic properties of oil/water emulsions with      dipalmitoylphosphatidylcholine, Colloids and Surfaces A: Physicochemical and Engineering Aspects,  302 (2007) 141-149

Model studies on the n-alkane emulsions stability Progress in Colloid and Polymer Science, 1997, 105, pp. 260–267

Studies of oil-in-water emulsion stability in the presence of new dicephalic saccharide-derived surfactants , Colloids and Surfaces B: Biointerfaces, 25 (2002) 243-256

Line 225

This sentence is too generalized, it's hard to consider -15 mV close to 0, right?

Line 234

You have some potential explanation for such a low value of fit in one case, the others had values ​​above 0.9?

Line 256-257

In my opinion these phrases are considered colloquial, give them a more substantive, more chemical one.

Figure 4

Latin names, regardless of whether they are in the main text, in the drawing or in the bibliography, should be spelled correctly in accordance with the nomenclature, i.e. italics.

In addition, Figures 4 and 5 differ significantly in the number of measurement points, why is it possible to standardize it?

Line 271 onwards

This fragment should be the main text in larger type and not the continuation of the caption under the figure

Line 292 onwards

Sense of this sentence?, please English correct

Conclusion

This section should be more informative. Please explain in the Conclusions sections how this paper contributes to new fundamental understanding.

Bibliography

In vitro should be also Italics

Bibliography should be carefully improved, sometimes old citation. Besides, the citations style is not uniform, not always consistent with the recommendations of this journal.

I can recommend this article, but after revision.

Author Response

Dear Reviewer,

We appreciate your comments and suggestions to improve our work.
Please see the attached document with the responses to your comments.

Best regards

Reviewer 2 Report

The article „Comparison between Quinoa and Quillaja saponins in the formation, stability and digestability of astaxanthin – canola oil emulsions“ is focused on the preparation and characterization of o/w emulsions stabilized by natural saponins.

The idea of this manuscript is interisting and also meets the current trend of the use of natural and environmental friendly agents. But there are some issues in this paper which must be fixed before the potential publication.

line 56 – hydrolyzed

line 93 – specify o/w

line 93 – I think o/w 1% is a very low concentrated emulsion – Why so low concentration? Have you also tried to prepare any more concentrated o/w?

line 119 – Have you also measured ESI immediately after preaparation? Were all samples stables after preparation?

line 123 – When was this measurement provided?

line 173 + whole manuscript (line 175, 195, 197, 237, 241, 321, 347....etc.) – be careful: -29.57mV ...emulsions had a lower charge of -25.1mV – you are talking about the absolute value?!!! mathematically it is wrong ...

line 198 – Figure 1. – Have you prepared the same emulsions with the scale of Tween 20 concentration?

Figure 1.a – How would you explain this rapid increasing in particle size (quinoa emulsifier 5%)?

line 206 – significant

line 214 – Figure 2. – Which concentration of emulsifiers has been used? 1%?

Figure 2. is not mentioned in the text ...

line 241 – at 35°C

line 315 – I suggest to discuss the efficiency of used emulsifiers (correlation) – Tween 20 as a synthetic emulsifier has better efficiency (and the effort is to add as small an amount as possible) – so, the comparison of the same concentration (e.g. 1%) for Tween 20 and a natural alternative can be misleading

Author Response

Dear Reviewer,

Thank you very much for your comments and suggestions.
Please see the attached document with responses to your comments.

Best Regards.

Round 2

Reviewer 1 Report

Necessary corrections have been made.

Author Response

There are no new comments from the reviewer.
Changes have been made according to the suggestions of the Academic Editor.

Best regards.

Reviewer 2 Report

The authors adressed all requested changes and recommendations.

Author Response

(The authors gave the same response as above.)
